# High Levels of Antibiotic Resistance in MDR-Strong Biofilm-Forming *Salmonella* Typhimurium ST34 in Southern China

**DOI:** 10.3390/microorganisms11082005

**Published:** 2023-08-03

**Authors:** Yuan Gao, Kaifeng Chen, Runshan Lin, Xuebin Xu, Fengxiang Xu, Qijie Lin, Yaping Hu, Hongxia Zhang, Jianmin Zhang, Ming Liao, Xiaoyun Qu

**Affiliations:** 1National and Regional Joint Engineering Laboratory for Medicament of Zoonoses Prevention and Control, Guangdong Laboratory for Lingnan Modern Agriculture, Guangzhou 510642, China; gaoy0115@163.com (Y.G.);; 2Key Laboratory of Zoonoses, Ministry of Agriculture, College of Veterinary Medicine, South China Agricultural University, Guangzhou 510642, China; 3Key Laboratory of Zoonoses Prevention and Control of Guangdong Province, College of Veterinary Medicine, South China Agricultural University, Guangzhou 510642, China; 4Animal Infectious Diseases Laboratory, College of Veterinary Medicine, South China Agricultural University, Guangzhou 510642, China; 5Faculty of Health Sciences, University of Macau, Macau SAR 999078, China; 6CAS Key Laboratory of Pathogenic Microbiology & Immunology, Institute of Microbiology, Chinese Academy of Sciences, Beijing 100101, China; 7Department of Microbiology, Shanghai Municipal Centre for Disease Control and Prevention, Shanghai 200015, China; 8Key Laboratory of Livestock Disease Prevention of Guangdong Province (YDWS202204), Guangzhou 510642, China

**Keywords:** *S. typhimurium*, ST34, drug resistance, ACSSuT, biofilm

## Abstract

*Salmonella* enterica subsp. enterica serovar Typhimurium (*S*. *typhimurium*) is an important zoonotic pathogen with important public health significance. To understand *S. typhimurium*’s epidemiological characteristics in China, multi-locus sequence typing, biofilm-forming ability, antimicrobial susceptibility testing, and resistant genes of isolates from different regions and sources (human, food) were investigated. Among them, ST34 accounted for 82.4% (243/295), with ST19 ranking second (15.9%; 47/295). ST34 exhibited higher resistance levels than ST19 (*p* < 0.05). All colistin, carbapenem, and ciprofloxacin-resistant strains were ST34, as were most cephalosporin-resistant strains (88.9%; 32/36). Overall, 91.4% (222/243) ST34 isolates were shown to have multidrug resistance (MDR), while 53.2% (25/47) ST19 isolates were (*p* < 0.05). Notably, 97.8% (45/46) of the MDR-ACSSuT (resistance to Ampicillin, Chloramphenicol, Streptomycin, Sulfamethoxazole, and Tetracycline) isolates were ST34, among which 69.6% (32/46) of ST34 isolates were of human origin, while 30.4% (14/46) were derived from food (*p* < 0.05). Moreover, 88.48% (215/243) ST34 showed moderate to strong biofilm-forming ability compared with 10.9% (5/46) ST19 isolates (*p* < 0.01). This study revealed the emergence of high-level antibiotic resistance *S. typhimurium* ST34 with strong biofilm-forming ability, posing concerns for public health safety.

## 1. Introduction

*Salmonella* is a food-borne pathogen with a broad range of host tropism, causing more than 90 million cases of salmonellosis annually worldwide. There are over 2500 *Salmonella* serotypes identified around the world, among which *Salmonella enterica* subsp. *enterica* serovar Typhimurium (*S. typhimurium*) is one of the most prevalent and widely distributed in China, with numerous hosts [1]. Transmission through contaminated food, such as pork and poultry, is the main route of *S. typhimurium* human and animal infections. The mortality in relation to *S. typhimurium* is three times higher than that associated with other *Salmonella* serotypes [2]. Therefore, the dispersal of *S. typhimurium* deserves more attention.

Multi-locus sequence typing (MLST), based on bacterial housekeeping genes, is one of the methods used to investigate the genetic characteristics of *S. typhimurium* strains and further elucidate their transmission routes. A total of 63 *S. typhimurium* sequence types (STs) have been found globally, including ST34 (57%) and ST19 (28.4%) [3]. Various STs of *S. typhimurium* exhibit different levels of pathogenicity and drug resistance. According to an analysis of *S. typhimurium* based on Enterobase database, in recent years, the prevalence of different STs of *S. typhimurium* in the world has changed significantly, including the predominant STs [4,5,6]. Thus, it is crucial to examine the prevalence patterns of *S. typhimurium* STs for their effective prevention and control.

However, the prevention and control of *S. typhimurium* is challenging because of the rapid development of drug resistance caused by the overuse of antibiotics [7]. The emergence and spread of multidrug resistance (MDR, defined as resistance to more than three classes of drugs) in *S. typhimurium* pose a severe threat to humans via foodborne infection [8,9,10]. Significant regional disparities in the antibiotic resistance patterns of bacteria have been demonstrated in China [11]. The prevalence of multidrug-resistant *Salmonella* has increased dramatically over the past decade, with MDR-*S. typhimurium* posing therapeutic problems because of escalating morbidity and mortality [12]. Worse still, the MDR-ACSSuT resistance pattern (defined as resistance to Ampicillin, Chloramphenicol, Streptomycin, Sulfamethoxazole, and Tetracycline) is related to death rates that are 4.8 times greater than that from infection with non-MDR-ACSSuT strains [13]. In addition to drug resistance at the cellular level, *S. typhimurium* acquires adaptive resistance, such as biofilms. Studies have shown that the production of biofilms enhances the drug resistance of strains [14]. Additionally, several recent investigations have reported the emergence of virulent multidrug-resistant bacterial pathogens of different origins, increasing the necessity of the proper use of antibiotics as well as the application of rapid accurate diagnostic tools for the screening of emerging virulent MDR strains [15,16,17].

In Guangdong, Shanghai, China, *S. typhimurium* ranked first or second in recent years [18,19]. Previous research mainly focused on one province in China; therefore, typical southern provinces were chosen for this study. To acquire a greater understanding of *S. typhimurium* prevalence in China, it is necessary to investigate the relationship between drug resistance, biofilm formation capacities, and STs with diverse epidemic patterns. Therefore, in this study, we investigated the drug resistance and biofilm-forming capabilities of different STs of *S. typhimurium* collected between 2007 and 2017 from various regions and sources (human, food) in China.

## 2. Materials and Methods

### 2.1. Specimen Collection and Isolate Identification

The *Salmonella* strains used in this study were collected from five provinces, including Shanghai, Guangdong, Guangxi, Chongqing, and Sichuan from 2007 to 2017. Clinical Salmonella strains were provided by the provincial Centers for Disease Control and regional hospitals. Food and environmental samples were collected from a retail market and slaughterhouses. *Salmonella* was isolated according to the U.S. FDA Bacteriological Analytical Manual [20]. Isolates with typical *Salmonella* phenotypes were further confirmed using API identification kits (bioMérieux, Durham, NC, USA), and O and H antigens were characterized using slide agglutination with salmonella diagnostic serum (S & A Reagents Lab, Bangkok, Thailand). The serological determination of *Salmonella* serotypes was performed in accordance with the Kauffmann–White scheme [21].

### 2.2. Polymerase Chain Reaction (PCR) Amplification and Multi-Locus Sequence Typing (MLST)

Genomic DNA was isolated using the InstaGene Matrix (Bio-Rad, Hercules, CA, USA) according to the manufacturer’s protocol. All primer sequences for amplification and sequencing were obtained from the MLST Databases of the University of Warwick (www.mlst.warwick.ac.uk/mlst/dbs/Senterica, accessed on 21 June 2023). The PCR cycling conditions were as indicated in instructions posted on the website. The PCR products were purified using Sephadex G-50 fine resin (GE Healthcare Bio-Sciences AB, Uppsala, Sweden). Nucleotide cycle-sequencing was performed directly on purified PCR templates using automated Sanger dideoxy chain termination methods and the primers described on the MLST website. Sequences of seven housekeeping genes (*aroC*, *dnaN*, *hemD*, *hisD*, *purE*, *sucA*, and *thrA*) were compared with those in the MLST database (http://mlst.warwick.ac.uk/mlst/dbs/Senterica accessed on 21 June 2023) to obtain the allele number and sequence type (ST) number for each isolate. Sequence information for newly assigned alleles and STs was deposited in the MLST database [17].

### 2.3. Antimicrobial Susceptibility Testing

The detection of *S. typhimurium* susceptibility to antimicrobials was carried out using the Agar dilution method as described by the Clinical and Laboratory Standards Institute [22]. The following antibiotics (Oxoid, Basingstoke, UK) were used. β lactams antibiotics: ampicillin (AMP), cefepime (FEP), cefotaxime (CTX). Aminoglycosides antibiotics: gentamicin (GEN) and amikacin (AK). Fluroquinolone antibiotics: ciprofloxacin (CIP), ofloxacin (OFX), and nalidixic acid (NAL). Amphenicol antibiotics: chloramphenicol (C) and florfenicol (FFC). Carbapenems antibiotic: imipenem (IPM). Colistin antibiotic: polymyxin B (PB). Streptomycin antibiotic: streptomycin (STR). Sulfonamides antibiotic: sulfisoxazole (SUL). Tetracyclines antibiotic: tetracycline (TET). *Escherichia coli* ATCC 25,922 served as the control. The results of antibiotic sensitivity were assessed according to the CLSI criteria [22].

### 2.4. Detection of Biofilm Formation Ability

Biofilm production was assessed using a previously described protocol with some modifications [18]. Briefly, overnight bacterial cultures were grown at 37 °C with shaking and adjusted to give an optical density (OD570 nm) of 0.01 via dilution with (1:10) tryptic soy broth (TSB). A 100 μL volume of bacterial suspension was added to each well of a 96-well plate and incubated at 28 °C for 48 h without shaking. The medium was then discarded, and each well was washed with water and stained with 100 μL of crystal violet for 15 min. Finally, the wells were washed three times with water, and biofilm formation was assayed via the addition of 100 μL of 95% ethanol to measure the OD 570 nm. Each isolate was assayed in eight individual replicates. The negative controls contained only 100 μL of diluted TSB according to the data obtained in the present investigation and the suggested criteria [19]. Strains were classified into one of four categories: non-biofilm producers, weak biofilm producers, moderate biofilm producers, or strong biofilm producers.

### 2.5. Detection of Resistance Genes and Mutations

Chromosome fluoroquinolone regulatory mutations (*gyrA* and *parC*) were analyzed using PCR and sequencing. Plasmid-mediated resistance genes, including *oqxAB*, *qepA*, *aac(6′)-Ib*, *qnrA, qnrB*, *qnrC*, *qnrD*, and *qnrS* (fluoroquinolone resistance genes); *bla_SHV_*, *bla_NDM-5_*, *bla_OXA-1_*, *bla_CTX m_* group, and *bla_OXA_
*(β-lactamase genes); and *mcr-1* (colistin resistant gene) were analyzed using PCR [23]. The total DNA of *Salmonella* was used as a template for PCR amplification, and the amplification products were detected using agarose gel electrophoresis.

### 2.6. Statistical Analysis

The comparison of frequencies was calculated by using the chi-squared test using SAS 9.2 (SAS Institute, Cary, NC, USA). A *p*-value < 0.05 was considered statistically significant.

## 3. Results

### 3.1. S. typhimurium Isolation and Identification

A total of 295 *S. typhimurium* strains were obtained from five provinces, including Shanghai (124), Guangdong (72), Guangxi (29), Chongqing (45), and Sichuan (25) from 2007 to 2017. Of the isolates, 54.9% (162/295) were isolated from patients, and 45.1% (133/295) were isolated from animal and foods samples (mostly from pork or pork products (63.9%; 85/133)).

### 3.2. MLST Analysis

A total of 243 (243/295; 82.4%) isolates belonged to genotype ST34, and 47 (47/295; 15.9%) isolates were ST19 (Table 1). The remaining five isolates were of human origin and belonged to ST36, ST99, and ST1557, respectively. For the clinical isolates, ST34 accounted for 81.5% (132/162), followed by ST19 (25/162; 15.4%). For the food isolates, ST34 (111/133; 83.5%) was the dominant MLST type, followed by ST19 (22/133; 16.5%). Except ST36, all the isolates belonged to eBG1 (eBrust Group).

### 3.3. Antimicrobial Susceptibility Testing and Distribution of MDR

The isolates showed high antimicrobial resistance to SUL (253/295; 85.8%) and TET (226/295; 76.6%), followed by florfenicol (204/295; 69.2%), AMP (201/295; 68.1%), NAL (196/295; 66.4%), C (149/295; 50.5%), and STR (146/295; 49.5%) (Appendix A). In addition, the rates of resistance to CIP, CTX, and FEP were18.6% (55/295), 119% (35/295), and 1.4% (4/295), respectively. Surprisingly, 3.1% (9/295) of the isolates and 0.3% (1/295) of the isolates were resistant to PB and IPM, respectively. All the strains were susceptible to amikacin. Notably, ST34 strains exhibited stronger drug-resistant characteristics than ST19, particularly toward AMP, NAL, CIP, OFX, FFC, STR, SUL, and TET (Appendix A). In addition, the isolates that exhibited resistance to CTX and PB were all ST34s. Overall, 88.9% (32/36) of the isolates that were resistant to cephalosporins were ST34s. Meanwhile, human and food origin strains displayed significantly different drug resistance characteristics, especially for AMP, CTX, STR, CIP, OFX, C, and TET (*p* < 0.05) (Figure 1). For the clinical isolates, 18.5% (30/162) of the them were resistant to the cephem class (FEP and CTX), while only 3.8% (5/133) of the food-derived strains were resistant to FEP and CTX. The food source isolates were mostly resistant to AMP (49.6%), NAL (66.9%), FFC (69.17%), STR (57.1%), SUL (84.2%), and TET (92.5%).

More importantly, 84.4% (249/295) of the isolates had developed multi-drug resistance (resistant to at least three classes of antibiotics, MDR ≥ 3), and 33 (33/295; 11.2%) isolates were resistant to seven classes of antibiotics (Table 2). The majority of the ST34 isolates (222/243; 91.3%) were MDR strains, in contrast to only 53.2% (25/47) of the ST19 isolates. Moreover, 16.2% (41/253) of the ST34 strains displayed MDR ≥ 6, in contrast to only 2.1% (1/47) of ST19 strains. Six ST34 strains were resistant to seven classes of antibiotics. In addition, 69.6% (32/46) of isolates from humans displayed the ACSSuT resistance pattern, in contrast to only 30.4% (14/46) of isolates from food–animal sources (*p* < 0.01). Notably, 45 out of 46 (97.8%) MDR-ACSSuT *S. typhimurium* isolates were identified as ST34 (Appendix A). Among them, all the MDR-ACSSuT *S. typhimurium* displayed the ‘ACSSuT + FFC’ pattern. In addition, 93.4% (43/46) of strains displayed the ‘ACSSuT + NAL + FFC’ pattern and 41.3% (19/46) strains displayed the ‘ACSSuT + CIP + FFC + NAL + X’ (the number of X ≥ 0) drug resistance pattern. More importantly, 8.7% (4/46) displayed the ACSSuT + CIP + CTX + FFC + NAL + X’ (the number of X ≥ 0) pattern, which were co resistant to ciprofloxacin and cephalosporins. Furthermore, 13.0% (6/46) of ACSSuT isolates exhibited intermediate resistance to ciprofloxacin (minimum inhibitory concentration (MIC) = 2 μg/mL). In addition, 53.1% (17/32) of the human-derived ACSSuT strains were isolated from children under 2 years old, leading to severe diarrhea. Meanwhile, 56.3% (18/32) of them tended to occur in summer (June to August). Moreover, 71.4% (10/14) of the animal-derived ACSSuT strains were isolated from pork. 

### 3.4. Biofilm Formation Ability

A total of 273 (273/295; 92.5%) isolates produced biofilms, among which 220 (220/295; 74.6%) isolates were categorized as moderate or strong biofilm producers. Specifically, 93.3% (125/134) of the human-derived isolates showed moderate and strong biofilm formation capacity, while only 82.6% (90/109) of the food-derived isolates showed moderate and strong biofilm formation capacity. The ST34 isolates of human origin had a stronger ability to form biofilms than those of food origin (Table 3). Furthermore, ST34 and ST19 displayed very different characteristics with respect to biofilm formation ability; all the ST34 isolates formed biofilms, while 38.3% (18/47) of the ST19 strains were non-biofilm producers. In addition, 215 (88.5%) ST34 isolates were classified as moderate or strong biofilm producers, in contrast to only 5 (10.6%) ST19 isolates (*p* < 0.01). 

### 3.5. Detection of Antimicrobial Resistance Genes and Mutations

The results showed that all the ciprofloxacin-resistant isolates were ST34 with single mutations. Among them, 98.2% (55/56) contained a *gyrA* mutation. Codon 87 (94.5%; 52/55) was the main *gyrA* mutation observed in mutant isolates. Consequently, the most common mutation was D87N (46/56; 82.1%), followed by D87Y (6/56; 10.7%). Other *gyrA* mutations, such as S83F (3/56; 5.4%), were observed in only three isolates. Interestingly, there was no mutation in *parC*.

Four types of PMQRs (Plasmid-Mediated Quinolone Resistance genes) were detected, including *oqxAB* (49/56; 87.5%), *aac(6′)-Ib-cr* (45/56; 80.3%), *qnrB* (14/56; 37.5%), and *qepA* (0/56; 0%). Interestingly, the D87N (20/31) mutation strain trended to possess three PMQR genes—*oqxAB*, *aac(6′)-Ib-cr*, and *qnrB*—compared with the D87Y (1/10) strain (*p* < 0.01). Notably, *qnrB* was only detected in human-derived strains (55.3%; 21/38) and not in food-derived isolates.

In addition, the majority of (91.67%; 33/36) cephalosporin-resistant strains were ST34 isolates, of which 66.7% (24/36) were positive for ESBLs genes, including *bla_TEM_* (41.7%; 15/36), *bla_CTX-M_
*(10/36; 27.8%), and *bla_OXA_* (19.4%; 7/36). Three *bla_CTX-M_* subtypes were detected, including *bla_CTX-M-55_* (80%; 8/10), *bla_CTX-M-123_* (10%; 1/10), and *bla_CTX-M-3_
*(10%; 1/10), respectively (Table 4). In this study, *bla_SHV_*, *bla_PSE_*, *bla_VEB_*, and *bla_PER_* genes were not found among the isolates. 

In addition, all the colistin-resistant and carbapenem-resistant strains were ST34 isolates. Interestingly, 100% (9/9) of the PB-resistant strains harbored an *mcr-1* gene, and one carbapenem-resistant strain harbored the *bla_NDM-5_* gene.

## 4. Discussion

Non-typhoidal *Salmonella* enterica is one of the leading causes of food-borne diseases. In recent years, *S*. typhimurium has been extensively detected in patients and numerous types of foods across the globe, with a significant economic impact on food safety and public health [24].Therefore, to epidemiologically study *S. typhimurium*, we systematically determined the antimicrobial resistance and molecular characterization of *S. typhimurium* from various regions and sources (human, food) in China in order to examine the relationship between drug resistance, biofilm formation capacity, and STs with diverse epidemic patterns. 

To better understand the prevalence of *S. typhimurium*, MLST was used for bacterial genotyping, which demonstrated that ST34 was the most widespread MLST type from human and animal–food sources in China, followed by ST19. ST34 differs from ST19 by only one allele *(dnaN*), in which a single base change occurs. The majority of *S. typhimurium* strains from several provinces in China were probably from the same founder (complex one, eG1). This was consistent with previous reports in China but totally different from that in India [6,14]. More importantly, ST34 and ST19 are the most common STs of *S.* I 4, [5],12:i:-, which is a monophasic variant of serovar Typhimurium whose prevalence has increased in the United States and China and whose pathogenicity is higher than past *S. typhimurium* [25,26]. Our results revealed that, in the past, the majority of *S. typhimurium* originated from ST19; however. recently, ST34 has replaced ST19 as the predominant strain.

To explore the biological characteristics of these strains and the underlying factors involved in their prevalence and spread, we analyzed the tolerance of different strains to antimicrobial factors. *S. typhimurium* showed high resistance to traditional antibiotics such as sulfisoxazole, tetracycline, florfenicol, ampicillin, nalidixic acid, and chloramphenicol. Worryingly, 18.64%, 11.86%, and 3.1% of the isolates in our study were resistant to ciprofloxacin, cefotaxime, and polymyxin B, respectively, significantly greater than the proportions of resistant pathogens reported in previous studies [3]. Notably, resistance to CIP, FFC, and NAL has been maintained at a higher level among *S. typhimurium* clinical isolates from China compared with those from other countries, which might reflect the paucity of data regarding strict adherence to dosages [27,28]. Surprisingly, an imipenem-resistant strain was identified, demonstrating that the imipenem resistance gene bla_NDM-5_ is mediated by a transferable plasmid [29]. Although the use of antimicrobials is not generally recommended to treat salmonellosis, fluoroquinolones and extended-spectrum β-lactams might have been commonly applied to treat salmonellosis infections in China. Thus, the prevalence and drug resistance data suggest that the bacterium has spread resistance genes to food and animal-derived products in circulation.

In our study, the isolates from patients displayed more resistance to AMP, CTX, CIP, and C antibiotics than the isolates from food. However, for the antibiotics STR, OFX, and TET, the food-derived strains were more resistant (*p* < 0.05). The phenomenon of differently derived strains displaying different drug resistance patterns might be caused by the different dominant STs, which indicate that different origins of *S. typhimurium* have their own resistant characteristics [3]. Notably, even though the food-derived strains had a lower drug resistance percentage (3.8%) for FEP and CTX than the clinical strains (18.5%), the dispersion of drug resistance to third- and fourth-generation cephalosporins in food and animals requires great attention. Additionally, the emergence of high-level drug resistant strains in food and humans indicated that the drug resistance of *S. typhimurium* might spread in food transportation and circulation; the strict management of veterinary medicine and human medicine should be carried out to prevent mixed use. 

Based on this, we conducted a drug resistance analysis on different STs of *S. typhimurium*. As expected, ST34 was resistant to more antibiotic classes than ST19. Among them, the resistance rate of ST34 *S. typhimurium* was much higher than that in previous reports and was higher than that of the ST19 strains in this study [27]. In addition to traditional antibiotics, all the ciprofloxacin-resistant strains were ST34, and the majority (88.9%, 32/36) of cephalosporin-resistant strains were ST34. This phenomenon might be related to the epidemiology of ST34, clinical antimicrobial treatment, and management performances. Antibiotics that are used frequently and indiscriminately will positively select resistant isolates, possibly resulting in more difficulties with respect to clinical treatment. We recommend more careful surveillance with respect to different STs and antibiotics in China. In addition to resistance at the cellular level, adaptive resistance is also crucial. It is suspected that biofilm-associated infections enhance persistence and resistance to environmental stress, including antimicrobials [30]. Comparative investigations on the biofilm-forming ability of *S. typhimurium* from various sources and strains in China are limited. 

According to our results, the capacity to form biofilms correlated with ST. Compared with other STs, ST34 was more likely to develop a biofilm, with 88.5% of isolates exhibiting moderate biofilm capacity or above, while only 10.6% of ST19 isolates showed such a biofilm capacity (*p* < 0.05). In comparison to a previous investigation, the proportion of ST34 biofilm development increased considerably, though further investigation is required to pinpoint the cause [3]. The formation of biofilms generates an extracellular matrix comprising cellulose and coils and provides protection for bacterial survival, which might account for the higher resistance rate of ST34 *S. typhimurium* [31].

Notably, 93.0% of ST34 demonstrated MDR phenotypes compared with 48.9% of ST19 [9,19], which was much higher than the reported MDR rate in *S. typhimurium* (56.58%) [32]. Additionally, 48.1% of the ST34 strains showed MDR ≥ six classes of antibiotics, while only 2.1% of the ST19 strains showed such resistance. Alarmingly, 13.2% of the ST34 strains were resistant to seven classes of drugs, which shows their potential to develop into XDR (Extensively Drug-Resistant) strains. Meanwhile, 97.8%(45/46) of the MDR-ACSSuT *S. typhimurium* isolates were identified as ST34 in our study, which was much higher than the 14.5% reported previously (Appendix A) [23]. Notably, all the ACSSuT strains were resistant to florfenicol, which is a broad-spectrum antimicrobial of the chloramphenicol class for veterinary use in China. This might indicate that there is drug-resistant cross-transmission from animals to humans. In addition, quinolones and third-generation cephalosporins are considered as alternative first-line drugs to treat *Salmonella* infections [26]. Co-resistance to fluoroquinolones and third-generation cephalosporins is already a major public health problem. Unfortunately, 41.3% of the strains were co-resistant to quinolone and cephalosporins. It is possible that human-derived ACSSuT strains could cause severe diarrhea, particularly in children without a mature immune system. Notably, all strains displaying the ACSSuT pattern were strong biofilm producers, enhancing the bacteria’s adaptability and aiding the dissemination of drug resistance. Hence, the monitoring of these severe MDR-ACSSuT ST34 among *S.* Typhimurium should be strengthened to prevent their spread in China.

To facilitate a greater understanding of the fundamental factors underlying bacterial antimicrobial resistance and establish measures for its prevention, an examination of the molecular mechanisms responsible is required. For quinolone- and cephalosporin-resistant strains, their resistance is mainly attributed to mutations of quinolone resistance-determining regions (QRDRs), the plasmid-mediated quinolone resistance (PMQR) mechanism, and extended-spectrum β-Lactamases (ESBLs) in *Salmonella* [25,26]. Ciprofloxacin-resistant *Salmonella* strains usually have at least two *gyrA* gene mutations, along with mutations in the other topoisomerase genes [33]. However, in this study, for the ciprofloxacin-resistant ST34 strains, the majority of mutations focused on the single mutation of *gyrA* (98.2%; 55/56) rather than *parC* (0%; 0/56), which was totally different than that the *S.* Typhimurium isolated from the Henan province, which have more than two mutations in *gyrA* and *parC* [34]. Interestingly, there was no mutation in the *parC* gene, whereas *Salmonella enterica* subspecies *enterica* serovar Corvallis tend to have *parC* to confer a high level of ciprofloxacin resistance [35]. Meanwhile, the QRDR mutations were different from those of *Salmonella enterica* subspecies *enterica* serovar Enteritidis, where D87N (82.1%) was the dominant mutation rather than D87G, which was the dominant mutation in 55.45% of the *S.* Enteritis isolates, possibly reflecting the different characteristics of different strains in different regions [23]. Although PMQR genes mediate low-level resistance to quinolones, they can complement other mechanisms to promote the accumulation of target-site mutations and facilitate the selection of higher-level resistance. The results showed that *oqxAB* and *acc6-(lb’)-cr* accounted for 87.5% and 80.3% of PMQR genes, respectively, followed by *qnrB* (37.5%; 21/56), wherein the percentage of PMQR carriers was much higher than reported in a previous study [34]. The high proportion of QRDRs and PMQRs pose a potential threat because there will be an increased resistance to ciprofloxacin due to a concomitant increase in the selection of mutants [36]. 

We identified ESBL genes in the cephem-resistant isolates. Four kinds of β-lactamase resistance genes were detected. Among them, the main resistant genes were *bla_TEM-1_* and *bla_CTX-M_*, both of which are often plasmid-borne and easily transferred, thus leading to the horizontal spread of antimicrobial resistance. Notably, in our data, 80% (8/10) of the *bla_CTX-M_* genotypes were *bla_CTX-M-55_*, which belongs to the CTX-M-1 group, exhibiting high hydrolytic activity toward cefotaxime and ceftazidime [37]. Recently, a significant and increasing prevalence of *bla_CTX-M-55_*-producing bacteria in China has been detected in both animal and human populations, leading to an increase in cefepime resistance in *Salmonella* [38,39]. Additionally, *bla_NDM-5_*, a rare resistance gene in *Salmonella,* was detected and identified as a plasmid-encoded gene in *Salmonella* in our previous study [29]. Such a high proportion of resistant strains and genes suggests that antibiotics should be used with caution. 

Colistin is recognized as the last-resort antimicrobial agent for treating carbapenem-resistant Enterobacteriaceae (CRE) infections [40]. *Mcr-1*-mediated colistin resistance in bacteria is concerning because colistin is used to treat multidrug-resistant bacterial infections. The current study found that *mcr-1* was present in nine colistin-resistant ST34s, suggesting that the *mcr-1* gene was prevalent in humans and food sources before 2016. Among children with diarrhea, those younger than 8 years old had the highest prevalence of human-derived bacterial *mcr-1* (5/6), which was supported by a previous study [41]. In China, colistin has been banned from being used in animal feed. Despite this, we discovered that three out of nine (33.3%) *mcr-1*-containg ST34s were of food origin from Guangdong, which might be attributed to the plasmid-mediated horizontal transmission of the *mcr-1* gene. Therefore, colistin-resistant *Salmonella* requires better detection and careful surveillance.

In summary, among the *S. typhimurium* isolates, ST34 displayed high-level antibiotic resistance and MDR-strong biofilm-forming ability compare with ST19. Efforts for the prevention and management of *S. typhimurium* should consider the multicellular behavior of the phenotypes of such strains.

## 5. Conclusions

This study aimed to investigate the drug resistance and biofilm-forming capabilities of different STs of *S. typhimurium* isolated from human, food, and animal sources in China. Our findings revealed that ST34 was the dominant ST, exhibiting significant drug resistance to traditional and defense-line drugs compared to ST19. Furthermore, we demonstrated that ST34 possesses a higher propensity for biofilm formation, potentially enhancing its adaptability in clinical and food-related infections. Therefore, based on the high levels of antibiotic resistance emerging in MDR-strong biofilm-forming *S. typhimurium* ST34 in southern China, our research sheds light on the potential challenges in combating infections and highlights the urgent need for targeted intervention strategies.

## Figures and Tables

**Figure 1 microorganisms-11-02005-f001:**
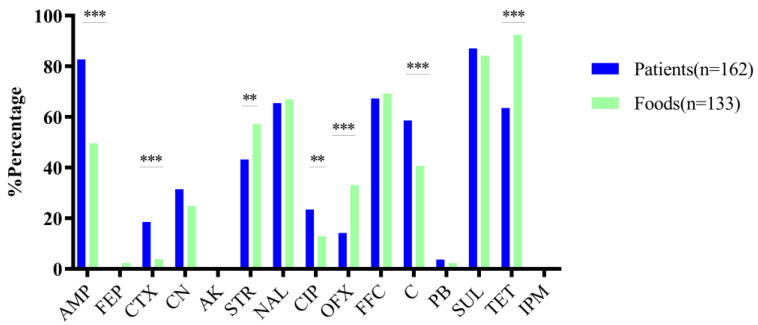
Drug resistance of *S.* Typhimurium of different origins. The isolates derived from patients were significantly resistant to AMP, C, CIP, and CTX. The food-derived isolates are more resistant to STR, OFX, and TET. The *y*-axis shows the resistance percentage for different antibiotics. *** *p* < 0.01; ** *p* < 0.05. AMP, Ampicillin; Sulfamethoxazole; C, Chloramphenicol; CIP, Ciprofloxacin; CTX, Cefotaxime; STR, Streptomycin; OFX, Ofloxacin; and TET, Tetracycline.

**Table 1 microorganisms-11-02005-t001:** MLST sequence typing of 295 *S.* typhimurium isolates.

STs	eBG	Sources	Isolates Number	Percentage
ST34	1	Human (132), food (111)	243	82.4%
ST19	1	Human (25), food (22)	47	15.9%
ST36	138	Human (2)	2	0.7%
ST99	1	Human (1)	1	0.3%
ST1557	1	Human (2)	2	0.7%

**Table 2 microorganisms-11-02005-t002:** Resistance to multiple antibiotic classes of the 295 *S. typhimurium* isolates.

Number of Antibiotic Classes They Exhibited Resistance to	Sources							Total
	Food (%)		Human (%)					
	ST19	ST34	ST19	ST34	ST36	ST99	ST1557	
0	3 (13.64%)	0 (0.00%)	5 (20.00%)	0 (0.00%)	0	1	0	9 (3.05%)
1	7 (31.82%)	4 (3.60%)	3 (12.00%)	3 (2.27%)	2	0	2	21 (7.12%)
2	3 (13.64%)	3 (2.70%)	3 (12.00%)	7 (5.30%)	0	0	0	16 (5.42%)
≥3	9 (40.91%)	105 (94.59%)	14 (56.00%)	121 (91.67%)	0	0	0	249 (84.41%)
3	5 (22.73%)	15 (13.51%)	4 (16.00%)	22 (16.67%)	0	0	0	46 (15.59%)
4	2 (9.09%)	16 (14.41%)	6 (24.00%)	12 (9.09%)	0	0	0	36 (12.20%)
5	2 (9.09%)	19 (17.12%)	3 (12.00%)	25 (18.94%)	0	0	0	49 (16.61%)
6	0 (0.00%)	41 (36.94%)	0 (0.00%)	44 (33.33%)	0	0	0	85 (28.81%)
7	0 (0.00%)	14 (12.61%)	1 (4.00%)	18 (13.64%)	0	0	0	33 (11.19%)
Total	22 (100.00%)	111 (100.00%)	25 (100.00%)	132 (100.00%)	2	1	2	295 (100.00%)

**Table 3 microorganisms-11-02005-t003:** Biofilm formation capacities of the total 295 *S.* Typhimurium isolates.

STs		ST34		ST19		ST36	ST99	ST1557	Total
Sources		Food	Human	Food	Human	Human			
Biofilm FormationAbility	None	0 (0.00%)	0 (0.00%)	6 (27.27%)	12 (48.00%)	1	1	2	22 (7.46%)
	Weak	19 (17.43%)	9 (6.72%)	14 (63.64%)	10 (40.00%)				273 (92.54%)
	Moderate	8 (7.34%)	30 (22.39%)	2 (9.09%)	2 (8.00%)	1			
	Strong	82 (75.23%)	95 (70.90%)	0 (0.00%)	1 (4.00%)				
	Total	109	134	22	25	2	1	2	295

**Table 4 microorganisms-11-02005-t004:** The antimicrobial-resistant genes of *S.* Typhimurium isolates.

Resistance Classification	Genes	Mutation	Proportion (%)
PMQR (plasmid-mediated quinolone resistance)	*qnrA*		0.0% (0/56)
	*qnrB*		37.5% (21/56)
	*qnrC*		0.0% (0/56)
	*qnrD*		0.0% (0/56)
	*qnrS*		0.0% (0/56)
	*acc6-(lb’)-cr*		80.3% (45/56)
	*oqxAB*		87.5% (49/56)
	*qepA*		0% (0/56)
QRDR (quinolone resistance-determining region)	*gyrA*	D(Asp)87N(Asn)	82.1% (46/56)
		D(Asp)87Y(Tyr)	10.7% (6/56)
		S(Ser)83F(Phe)	5.4% (3/56)
		No mutation	0.2% (1/56)
	*parC*	No mutation	100.0% (56/56)
β-lactams	*bla_CTX-M_*		27.8% (10/36)
	*bla_TEM_*		41.7% (15/36)
	*bla_OXA_*		19.4% (7/36)
Carbapenem	*bla_NDM-5_*		100% (1/1)
Colistin	*mcr-1*		100% (9/9)

## Data Availability

The data presented in this study are available upon request from the corresponding author.

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
