# Peer review of "High Levels of Antibiotic Resistance in MDR-Strong Biofilm-Forming Salmonella Typhimurium ST34 in Southern China"

_microorganisms, 2023, doi:10.3390/microorganisms11082005_

Round 1

Reviewer 1 Report

overall, it is an interesting study revealing the emergence of high level MDR S. Typhimurium ST34 in Southern China.

there are few comments;

in the methods section (lines 84-88), authors need to include details on strains of Salmonella Typhimurium; for example, how many clinical isolates and their source (is it feces/blood)- also, how many strains of food isolates and their source?

authors need to take care of writing bacterial name correctly as serovar (Tyhimurium) should start with  a capital (T) and it is not written in italics while genus name (Salmonella) is written in italics.

there are few typos including symbols for degrees (lines 114, 117, ..) and ul (lines 116, ...)

Author Response

Comment1: In the methods section (lines 84-88), authors need to include details on strains of Salmonella Typhimurium; for example, how many clinical isolates and their source (is it feces/blood)- also, how many strains of food isolates and their source?

Response 1: Thanks for your comments. I’ve added the details, where the detailed information strains are shown in the result part (3.1 Typhimurium isolation and identification).

Comment 2: Authors need to take care of writing bacterial name correctly as serovar (Typhimurium) should start with a capital (T) and it is not written in italics while genus name (Salmonella) is written in italics.

Response 2: Thanks for your comments. I’ve revised all the Typhimurium name into capital (T), and all the genus name (Salmonella) is written in italics.

Comment 3: there are few types including symbols for degrees (lines 114, 117, ..) and underlines 116, ...)

 Response 3: Thanks for your comments. I’ve revised the symbols in line 114-117.

Reviewer 2 Report

In manuscript "Emerging high-levels of antibiotic resistance in MDR-strong biofilm forming Salmonella Typhimurium ST34 in Southern China" s systematic multi-regional study of Salmonella Typhimurium isolates were described. From the title I expected that Salmonella Typhimurium ST34 isolates would be analyzed in more detail, while in essence the paper is a characterization of the different collected isolates of this Salmonella serogroup. I am missing a statistical analysis that would link the obtained results of bacterial typing, the source of infection, resistance genes and the ability to form biofilms, which would improve the quality of the work. Was this isolate somehow special and in which region did it dominate?

Line 109 write the name of the bacteria in italics

There are a lot of mistakes in the manuscript and they need to be corrected.

 Moderate editing of English language required.

Author Response

Comments 1: In manuscript "Emerging high-levels of antibiotic resistance in MDR-strong biofilm forming Salmonella Typhimurium ST34 in Southern China" s systematic multi-regional study of Salmonella Typhimurium isolates were described. From the title I expected that Salmonella Typhimurium ST34 isolates would be analyzed in more detail, while in essence the paper is a characterization of the different collected isolates of this Salmonella serogroup. I am missing a statistical analysis that would link the obtained results of bacterial typing, the source of infection, resistance genes and the ability to form biofilms, which would improve the quality of the work. Was this isolate somehow special and in which region did it dominate?

Response 1: Thanks for your comments.

This research aims to investigate the prevalence of S. Typhimurium, ST34 and ST19, isolated from human, food, and animal sources in various regions of Southern China. Our findings reveal that the contamination of S. Typhimurium in these regions is predominantly associated with the prevalence of ST34. In contrast to ST19, ST34 frequently harbors a higher number of drug-resistant genes and exhibits enhanced biofilm-forming capabilities, resulting in increased drug resistance. The significance of these biological traits in relation to ST genotype was validated through chi-squared tests. The following sections provide a detailed analysis of our findings.

We found several points to indicate the relationship between the ST type (ST34) and MDR-strong biofilm, which includes the multidrug resistance pattern, resistant genes and biofilm forming condition.

First, we found the dominant ST type of S. Typhimurium is ST34 in Southern China, particularly in certain regions (Guangdong, Guangxi, Chongqing, Shanghai, Sichuan). These regions occupy a large proportion of the area and population of southern China, as well as a large proportion of meat food consumption.

Second, ST34 strains exhibit higher drug resistance than ST19. Notably, all colistin, carbapenem, and ciprofloxacin-resistant strains were ST34, some of which are last-line drug in human clinical infections.

Third, much more MDR strains are ST34 (91.4%) compared to ST19 (53.2%). Significantly, 97.8%strains exhibit ACSSuT drug resistance patterns, and we classified the patterns into different groups. Detailed information is discussed in the result and discussion in lines 173-192 and in lines 308-325

Fourth, ST34 showed stronger biofilm-producing ability than ST19. All the ST34 isolates formed biofilm, while 38.7% (18/47) of ST19 strains were non-biofilm producers. In addition, 215 (88.48%, 215/243) ST34 isolates were classified as moderate or strong biofilm producer, in contrast to only 5 (10.6%) ST19 isolates (P < 0.01). (Detailed information was showed in 3.4. Biofilm formation ability), These findings may account for the higher drug resistance rate of ST34. We’ve made analyses and comparisons in the discussion.

Comments 2: Line 109 write the name of the bacteria in italics

Response 2: Thanks for your comments. I’ve revised the Escherichia coli in to italics Escherichia coli.

Comment 3: There are a lot of mistakes in the manuscript and they need to be corrected.

Response 3: Thanks for your comments. I’ve revised the mistakes. For example, I’ve revised the symbols in line 114-117. I’ve revised all the Typhimurium name into capital (T), and all the genus name (Salmonella) is written in italics.

Comment 4: Moderate editing of English language required.

Response 4: Thanks for your comments. I’ve edited the English language in several places in main text. For example, I’ve rephased the abstract to much more specifically. I’ve revised the names of bacterial pathogens and genes in the correct form. I’ve modified the introduction, methods and results, and rephrased the main conclusion of my findings (line70-74, 85-94,375-378, 380-388). I’ve added the full expression before the abbreviations.

Reviewer 3 Report

Comments to authors:

-The current study is interesting; however, the authors should address the following comments to improve the quality of the manuscript:

-The manuscript should be revised for English editing and grammar mistakes.

- Please write the scientific names of bacterial pathogens and genes in the correct form all over the manuscript and the references section.

Title:

I think the work would benefit from the title that contains the main conclusion of the study (should be derived from the conclusion). Please modify the title.

Abstract:

- The abstract must illustrate the used methods and the most prevalent results (give more hints about methods and results). Besides, rephrase the aim of the work and the main conclusion of your findings.

-A graphical abstract is recommended (If possible).

- Add the full expression before the abbreviations.

-Introduction: (it needs to be more informative):

-Give a hint about the virulence factors, the mechanism of disease occurrence, and infecions caused by Salmonella Typhimurium.

- The authors should illustrate the public health importance concerning the emergence of virulent multidrug-resistant (MDR) bacterial pathogens that reflect the necessity of application rapid specific diagnostic tools;  

Authors could add the following paragraph:

Multidrug resistance has been increased all over the world that is considered a public health threat. Several recent investigations reported the emergence of virulent multidrug-resistant bacterial pathogens from different origins that increase the necessity of the proper use of antibiotics as well as the application of rapid accurate diganostic tools for screening of the emerging virulent MDR strains. You are advised to cite the following valuable studies:

1. PMID: 34203245

2. PMID: 35971557

3. PMID: 36365013

-Rephrase the aim of the work to be clear and better sound.

Material and methods:

- Support all methods with updated specific references.

• Add the company, city, and country of the used chemicals and reagents.

- Add this subtitle: Isolation and identification of Salmonella Typhimurium: 

 - Illustrate the source of the tested strains.

Discuss in detail the methods of isolation and identification of Salmonella Typhimurium. Besides, specific references should be added.

- Disscuss in details serotyping of the recovered Salmonella Typhimurium isolates. 

-Antimicrobial susceptibility testing should be performed:

-Please, explain in detail

•Add the names of all tested antibiotics and the antimicrobial classes.

•The authors are advised to classify the tested isolates to MDR , XDR, and PDR as described by Magiorakos et al.

Magiorakos AP, Srinivasan A, Carey RB, Carmeli Y, Falagas ME, Giske CG, et al. Multidrug-resistant, extensively drug-resistant and pandrug-resistant bacteria: An international expert proposal for interim standard definitions for acquired resistance. Clin Microbiol Infect. 2012; 18:268–81. doi:10.1111/j.1469-0691.2011.03570.x.

- To increase the impact of the present study, the detection of virulence and antimicrobial resistance genes in the recovered Salmonella Typhimurium. should be performed. Afterwards, the correlation between phenotypic and genotypic multidrug resistance should be performed.

  -Results: (Good Presentation)

- Please add a starting paragraph to the results section to briefly introduce the topic, your goals and 

hypothesis and a short summary of what you did in this work.

-Add this subtitle: Phenotypic characteristics of the recovered Salmonella Typhimurium:

• Illustrate in detail the phenotypic characteristics of the recovered isolates.

-Antimicrobial susceptibility testing:

• -Illustrate in a new table the occurrence of MDR (Multidrug resistance) among the recovered isolates as the following (illustrate the names of the antimicrobial classes and different antibiotics):

No. of strains % Type of resistance

R, MDR, and XDR Phenotypic multidrug resistance

(Antimicrobial classes and different antibiotics). The antibiotic-resistance genes

-To increase the impact of the present study, the detection of virulence and antimicrobial resistance genes in the recovered Salmonella Typhimurium. should be performed. Afterwards, the correlation between phenotypic and genotypic multidrug resistance should be performed.

-Increase the resolution of all figures (must be 600 dpi).

-Discussion:

 The authors are advised to illustrate the real impact of their findings without repetition of results.

- Please illustrate the mechanism of action of different virulence determinants of Salmonella Typhimurium.

- Please illustrate the mechanism of antimicrobial resistance in Salmonella Typhimurium.

-Conclusion

- Should be rephrased to be sounded. A real conclusion should focus on the question or claim you articulated in your study, which resolution has been the main objective of your paper?

The manuscript should be revised for English editing and grammar mistakes.

- Please write the scientific names of bacterial pathogens and genes in the correct form all over the manuscript and the references section.

Author Response

Comments to authors:

-The current study is interesting; however, the authors should address the following comments to improve the quality of the manuscript:

-The manuscript should be revised for English editing and grammar mistakes.

 Response: Thanks for your comment. I’ve revised the English editing and grammar mistakes; For example, I’ve rephased the abstract to much more specifically. I’ve revised the names of bacterial pathogens and genes in the correct form. I’ve modified the introduction, methods and results, and rephrased the main conclusion of my findings (line70-74, 85-94,375-378, 380-388). I’ve added the full expression before the abbreviations.

- Please write the scientific names of bacterial pathogens and genes in the correct form all over the manuscript and the references section.

 Response: Thanks for your comment. I’ve revised the names of bacterial pathogens and genes in the correct form.

Title:

I think the work would benefit from the title that contains the main conclusion of the study (should be derived from the conclusion). Please modify the title.

Response: Thanks for your comment. We’ve modified the conclusion and rephased it, and modified the title into ‘High-levels of antibiotic resistance in MDR-strong biofilm forming Salmonella Typhimurium ST34 in Southern China.

Abstract:

- The abstract must illustrate the used methods and the most prevalent results (give more hints about methods and results). Besides, rephrase the aim of the work and the main conclusion of your findings.

 Response: Thanks for your comment. I’ve modified the methods and results, and rephrased the main conclusion of my findings in abstract.

-A graphical abstract is recommended (If possible).

 Response: Thanks for your comment. We’ve considered carefully your recommendation, but we consider that there are already graph and tables in our main text.

- Add the full expression before the abbreviations.

Response: Thanks for your comment. I’ve added the full expression before the abbreviations.

-Introduction: (it needs to be more informative):

-Give a hint about the virulence factors, the mechanism of disease occurrence, and infections caused by Salmonella Typhimurium.

Response: Thanks for your comment. I’ve revised the introduction properly.

- The authors should illustrate the public health importance concerning the emergence of virulent multidrug-resistant (MDR) bacterial pathogens that reflect the necessity of application rapid specific diagnostic tools;  

Authors could add the following paragraph:

Multidrug resistance has been increased all over the world that is considered a public health threat. Several recent investigations reported the emergence of virulent multidrug-resistant bacterial pathogens from different origins that increase the necessity of the proper use of antibiotics as well as the application of rapid accurate diganostic tools for screening of the emerging virulent MDR strains. You are advised to cite the following valuable studies:

  1. PMID: 34203245IF: 5.222 Q1
  2. PMID: 35971557IF: 4.177 Q2
  3. PMID: 36365013IF: 4.531 Q2

 Response: Thanks for your comment. After careful consideration of your suggestion, we think it is necessary to cite some diagnostic papers. So I’ve revised the introduction properly in the third paragraph in the introduction.

-Rephrase the aim of the work to be clear and better sound.

Response: Thanks for your comment. I’ve revise it properly (line 77-82).

Material and methods:

- Support all methods with updated specific references.

  • Add the company, city, and country of the used chemicals and reagents.

Response: Thanks for your comment. I’ve revise it properly and add more detailed information (line 90-94).

- Add this subtitle: Isolation and identification of Salmonella Typhimurium: 

 - Illustrate the source of the tested strains.

 Response: Thanks for your comment. I’ve revised it properly (line 85-94).

Discuss in detail the methods of isolation and identification of Salmonella Typhimurium. Besides, specific references should be added.

- Discuss in details serotyping of the recovered Salmonella Typhimurium isolates. 

-Antimicrobial susceptibility testing should be performed:

-Please, explain in detail

Response: Thanks for your comments. I’ve modified the methods part in 2.3 Antimicrobial susceptibility testing. And some specific references are cited (line 108-118). Meanwhile, I’ve summarized the isolates detail in the result 3.1.(The subtitle is modified as you suggested). And the antimicrobial susceptibility testing results are discussed in the result and discussion.

  • Add the names of all tested antibiotics and the antimicrobial classes.

Response: Thanks for your comment. I’ve revised it properly (line 110-119).

  • The authors are advised to classify the tested isolates to MDR , XDR, and PDR as described by Magiorakos et al.

Magiorakos AP, Srinivasan A, Carey RB, Carmeli Y, Falagas ME, Giske CG, et al. Multidrug-resistant, extensively drug-resistant and pandrug-resistant bacteria: An international expert proposal for interim standard definitions for acquired resistance. Clin Microbiol Infect. 2012; 18:268–81. doi:10.1111/j.1469-0691.2011.03570.xIF: 13.310 Q1 .

 Response: Thanks for your advice. I’ve cited this paper in the introduction part. Normally, in our study, MDR defined as resistant to more than three classes of antimicrobial agents. We’ve added the definition in the introduction. XDR defined as resistant to more than six classes of antimicrobial agents. No XDR/PDR strain have been found in our study.

- To increase the impact of the present study, the detection of virulence and antimicrobial resistance genes in the recovered Salmonella Typhimurium. should be performed. Afterwards, the correlation between phenotypic and genotypic multidrug resistance should be performed.

 Response: Thanks for your comment. I’ve chosen the strains that were resistant to ciprofloxacin order to better highlight the correlation between resistance and resistance phenotype. For those strains that are not resistant, we can leave them out to some extent because it will confuse the results of our work.

  -Results: (Good Presentation)

- Please add a starting paragraph to the results section to briefly introduce the topic, your goals and hypothesis and a short summary of what you did in this work.

Response: Thanks for your comment. I’ve added several sentences to briefly introduce the topic, goals, and hypothesis and short summary.

-Add this subtitle: Phenotypic characteristics of the recovered Salmonella Typhimurium:

  • Illustrate in detail the phenotypic characteristics of the recovered isolates.

 Response: Thanks for your comment. I've discussed the origin and ST typing of bacteria in the results section (3.1 and 3.2). and I’m not clear which phenotypes need to be discussed in the section of phenotype characteristics.

-Antimicrobial susceptibility testing:

  • -Illustrate in a new table the occurrence of MDR (Multidrug resistance) among the recovered isolates as the following (illustrate the names of the antimicrobial classes and different antibiotics):

No. of strains % Type of resistance

R, MDR, and XDR Phenotypic multidrug resistance

(Antimicrobial classes and different antibiotics). The antibiotic-resistance genes

 Response: Thanks for your comment. I’ve illustrated the occurrence of MDR strains in Table 2, and the resistance condition is shown in the supplemental tables.

-To increase the impact of the present study, the detection of virulence and antimicrobial resistance genes in the recovered Salmonella Typhimurium should be performed. Afterwards, the correlation between phenotypic and genotypic multidrug resistance should be performed.

Response: Thanks for your comment. We’ve considered it seriously before we design this research, but we think our study focused on the relationship between antibiotic resistance and biofilm in different S. Typhimurium (ST34 and ST19).

-Increase the resolution of all figures (must be 600 dpi).

 Response: Thanks for your comment. I’ve revised it in to 600dpi.

-Discussion:

 The authors are advised to illustrate the real impact of their findings without repetition of results.

- Please illustrate the mechanism of action of different virulence determinants of Salmonella Typhimurium.

Response: Thanks for your comment. We’ve considered it seriously before we design this research, but we think our study focused on the relationship between antibiotic resistance and biofilm in different S. Typhimurium (ST34 and ST19).

- Please illustrate the mechanism of antimicrobial resistance in Salmonella Typhimurium.

Response: Thanks for your comment. We’ve discussed and explained some mechanism of antimicrobial resistance in S. Typhimurium, such as PMQR-mediated drug resistance, ESBL-mediated drug resistance, mcr-1 mediated colistin resistance. We’ve explained the resistance phenotype in relation to its resistance mechanism.

-Conclusion

- Should be rephrased to be sounded. A real conclusion should focus on the question or claim you articulated in your study, which resolution has been the main objective of your paper?

Response: Thanks for your comment. I’ve rephrased the paragraph. The main objective in this study is to investigate drug resistance and biofilm-forming capabilities in S. Typhimurium from diverse sources in China.

Comments on the Quality of English Language

The manuscript should be revised for English editing and grammar mistakes.

- Please write the scientific names of bacterial pathogens and genes in the correct form all over the manuscript and the references section.

Response: Thanks for your comment. I’ve revised all the scientific name all over the manuscript and the references section.

Round 2

Reviewer 2 Report

I have no further comments.

Reviewer 3 Report

The authors have carried out significant changes to the manuscript. They have addressed most of the suggested corrections and comments. Really, it's an interesting study that has a significant impact. Now, the manuscript could be accepted.

Congratulations.